# Hydrothermal vents trigger massive phytoplankton blooms in the Southern Ocean

Mathieu Ardyna[1,2], Léo Lacour[1,3], Sara Sergi[4], Francesco d'Ovidio[4], Jean-Baptiste Sallée [4],
Mathieu Rembauville[1], Stéphane Blain[5], Alessandro Tagliabue[6], Reiner Schlitzer [7], Catherine Jeandel[8],
Kevin Robert Arrigo[2] & Hervé Claustre[1]

Hydrothermal activity is significant in regulating the dynamics of trace elements in the ocean. Biogeochemical models suggest that hydrothermal iron might play an important role in the iron-depleted Southern Ocean by enhancing the biological pump. However, the ability of this mechanism to affect large-scale biogeochemistry and the pathways by which hydrothermal iron reach the surface layer have not been observationally constrained. Here we present the first observational evidence of upwelled hydrothermally influenced deep waters stimulating massive phytoplankton blooms in the Southern Ocean. Captured by profiling floats, two blooms were observed in the vicinity of the Antarctic Circumpolar Current, downstream of active hydrothermal vents along the Southwest Indian Ridge. These hotspots of biological activity are supported by mixing of hydrothermally sourced iron stimulated by flow-topography interactions. Such findings reveal the important role of hydrothermal vents on surface biogeochemistry, potentially fueling local hotspot sinks for atmospheric $CO_2$ by enhancing the biological pump.

[1] Sorbonne Université & CNRS, Laboratoire d'Océanographie de Villefranche (LOV), 181 Chemin du Lazaret, F-06230 Villefranche-sur-mer, France.
[2] Department of Earth System Science, Stanford University, Stanford, CA 94305, USA. [3] Takuvik Joint International Laboratory, Laval University (Canada) - CNRS (France), Département de biologie et Québec-Océan, Université Laval, Québec, Québec G1V 0A6, Canada. [4] Sorbonne Université, CNRS, IRD, MNHN, Laboratoire d'Océanographie et du Climat: Expérimentations et Approches Numériques (LOCEAN-IPSL), F-75005 Paris, France. [5] Sorbonne Université & CNRS, Laboratoire d'Océanographie Microbienne (LOMIC), Observatoire Océanologique, F-66650 Banyuls/mer, France. [6] Department of Earth, Ocean and Ecological Sciences, School of Environmental Sciences, University of Liverpool, Liverpool, UK. [7] Alfred Wegener Institute, Helmholtz-Center for Polar- and Marine Research, Am Alten Hafen 26, 27568 Bremerhaven, Germany. [8] LEGOS (Université de Toulouse, CNRS, CNES, IRD, UPS), 14 avenue Edouard Belin, 31400 Toulouse, France. Correspondence and requests for materials should be addressed to M.A. (email: ardyna@stanford.edu)

Iron is an important resource limiting the efficiency of the biological pump over large areas of the global ocean[1–3]. Until recently, the role of hydrothermal activity in governing the ocean iron inventory and its effect on global biogeochemical cycles has been largely underestimated[4]. New discoveries regarding the distribution, number, type and activity of hydrothermal vent systems are forcing us to revisit existing paradigms related to hydrothermal vents and their impact on ocean biogeochemistry[2,4,5]. For example, a large plume of hydrothermal dissolved iron was recently observed extending several thousand kilometers westward from the southern East Pacific Rise across the South Pacific Ocean[6]. Based on these observations, estimates of the global hydrothermal dissolved iron input to the ocean interior has been increased to three to four gigamoles per year, which is more than fourfold higher than previous estimates[6].

Because the Southern Ocean (SO) is the largest iron-limited region of the global ocean, local phytoplankton are particularly sensitive to iron inputs[7,8]. The current paradigm holds that iron supplied from continental margins and sea ice drive hot spots of biological activity in the SO[8,9]. However, global model simulations indicate that the biological pump can be directly impacted by hydrothermal iron released along ridges within and outside the SO[10]. Winter mixing and/or upwelling brings these deep iron-enriched waters to the surface in the SO where, when coupled with intense lateral stirring, they may play an important role in fueling planktonic blooms over wide areas[11,12]. However, there is no observational evidence supporting the conclusion by models that hydrothermal iron is an important source enhancing biological activity. Here, we combine data from new autonomous platforms and satellite-derived observations to show that the upwelled hydrothermally influenced deep waters can indeed stimulate massive phytoplankton blooms in the Southern Ocean.

## Results

**Two unexpectedly large open ocean blooms.** By analyzing a circumpolar compilation of phytoplankton blooms captured by BGC-Argo floats (117 phytoplankton blooms, Fig. 1a), we observed two unexpectedly massive phytoplankton blooms in typically High Nutrient Low Chlorophyll (HNLC) waters of the SO (Fig. 1, red dots; see also the Supplementary Note for similar patterns based on particle backscattering). These phytoplankton blooms (in 2014 and 2015; Fig. 2) were observed in the Indian Sector of the SO (30–38°E, 48–55°S) using two phytoplankton biomass proxies: chlorophyll a (Chl a) and particle backscattering (Fig. 2b, c). The magnitude of these blooms (maximum depth-integrated biomass of 83.0 and 96.5 mg Chl a m$^{-2}$, respectively) is similar to those observed in highly iron-enriched waters downstream of the Crozet and Kerguelen Plateaus (mean 98.1 mg Chl a m$^{-2}$; Fig. 1, green dots) and in proximity to the sea-ice edge (mean 70.0 mg Chl a m$^{-2}$, purple dots), and more than twice that of HNLC waters of the SO (mean 42.0 mg Chl a m$^{-2}$, blue dots).

Such levels of phytoplankton biomass can only be achieved by significant iron enrichment. However, the location of these blooms is far from typical iron sources such as shallow continental shelves, melting sea-ice (Fig. 1a), and atmospheric dust deposition[13]. Therefore, the most plausible iron source for these blooms is upwelling from deep waters. Interestingly, these two massive blooms are in the vicinity of the eastward-flowing Antarctic Circumpolar Current (ACC), directly downstream of an arc of known active hydrothermal vents[14,15] (from 3517 to 4170 m deep; Fig. 2a, b), which should elevate deep ocean iron concentrations, and also directly downstream of the Southwest Indian Ridge (SWIR), which could promote vertical mixing and thus deliver these iron-enriched waters to the surface. This is supported by a suite of evidence indicating that deep-waters in

the region have a strong hydrothermal vent signature and are therefore likely to be iron-enriched (Fig. 3); flow-topography interactions in the region enhance turbulence and vertical advection, so that deep-waters are efficiently transported to the surface (Fig. 4); and once waters reach the surface, they are efficiently transported downstream into two branches feeding the regions where we observed massive phytoplankton blooms (Fig. 4).

**Hydrothermally influenced waters downstream of the SWIR.** Hydrothermal vents along spreading ocean ridges release large amounts of primordial He originating from the Earth's mantle, which is associated with high $\delta^3$He isotopic signature. $\delta^3$He is a conservative tracer and is commonly used to detect the presence of hydrothermal vent fluids in oceanic waters[6,16]. It has also been shown that hydrothermal vents are associated with elevated concentrations of iron, resulting in a tight covariance with He[6,16,17]. In the only pseudo-Lagrangian study of dispersal of iron from a hydrothermal plume, a constant relationship between dissolved iron concentration and $\delta^3$He was observed[6].

Two vertical sections of $\delta^3$He in the vicinity of the sampled blooms demonstrate a strong signature of hydrothermal vent activity in the waters of the SWIR region, which would imply iron-rich deep waters (Fig. 3). Along a meridional section at 30°E that crosses the SWIR, the high level of $\delta^3$He in waters above ~2000 m (10–12%) clearly indicates a hydrothermal signal between ~2000 m and the permanent thermocline (Fig. 3c). Note that $\delta^3$He drops above the permanent thermocline due to atmospheric exchange. Furthermore, a second section along ~55°S highlights the zonal variability in $\delta^3$He, whereby the elevated $\delta^3$He signature in deep water is only present downstream (east) of the SWIR (~25°E; Fig. 3b). These $\delta^3$He data clearly indicate that the arc of hydrothermal vents along the SWIR have a widespread influence on downstream iron release into waters between 2000 m and the permanent thermocline.

**Mixing by flow-topography interactions.** Furthermore, the dynamics associated with flow-topography interactions at the SWIR suggest that these hydrothermally-enriched deep waters are transported efficiently to the surface. Downstream of the SWIR, the flow is steered by topography into two branches associated with elevated eddy kinetic energy (EKE; one eastward branch around 50°S, and one southward branch around 35°E; Figs 3, 4a, b). Note that the trajectories of the two BGC-Argo floats (that drift at 1000 m) that recorded the two large blooms followed these two high EKE branches. This topographic flow increases EKE throughout the water column directly downstream, which likely enhances cross-stream buoyancy flux, whose vertical divergence is related to the upward transport of along-stream momentum[12,18]. In other words, deep waters are upwelled along the ACC branches downstream of the SWIR in the region of elevated EKE.

High-resolution numerical simulations of the dynamics of flow-topography interactions in the region[12,18] also strongly suggest that the ACC interacts with the SWIR. These numerical studies[12,18] indeed highlight two signatures of these dynamics and the associated deep-water upwelling, including a deep enhancement of EKE and an along-stream shallowing of isopycnals at depth. Consistent with these simulations, we find that the study region is associated both with elevated EKE at depth (derived from Argo trajectory-based velocities, Fig. 4b; see the Methods for more details) and a large along-stream vertical displacement of isopycnals (shallowing; up to an increase of 0.2 kg m$^{-3}$ at 750 m in the region between 28 and 38°E: Fig. 4c, d; see the Methods for more details). Such observational evidence

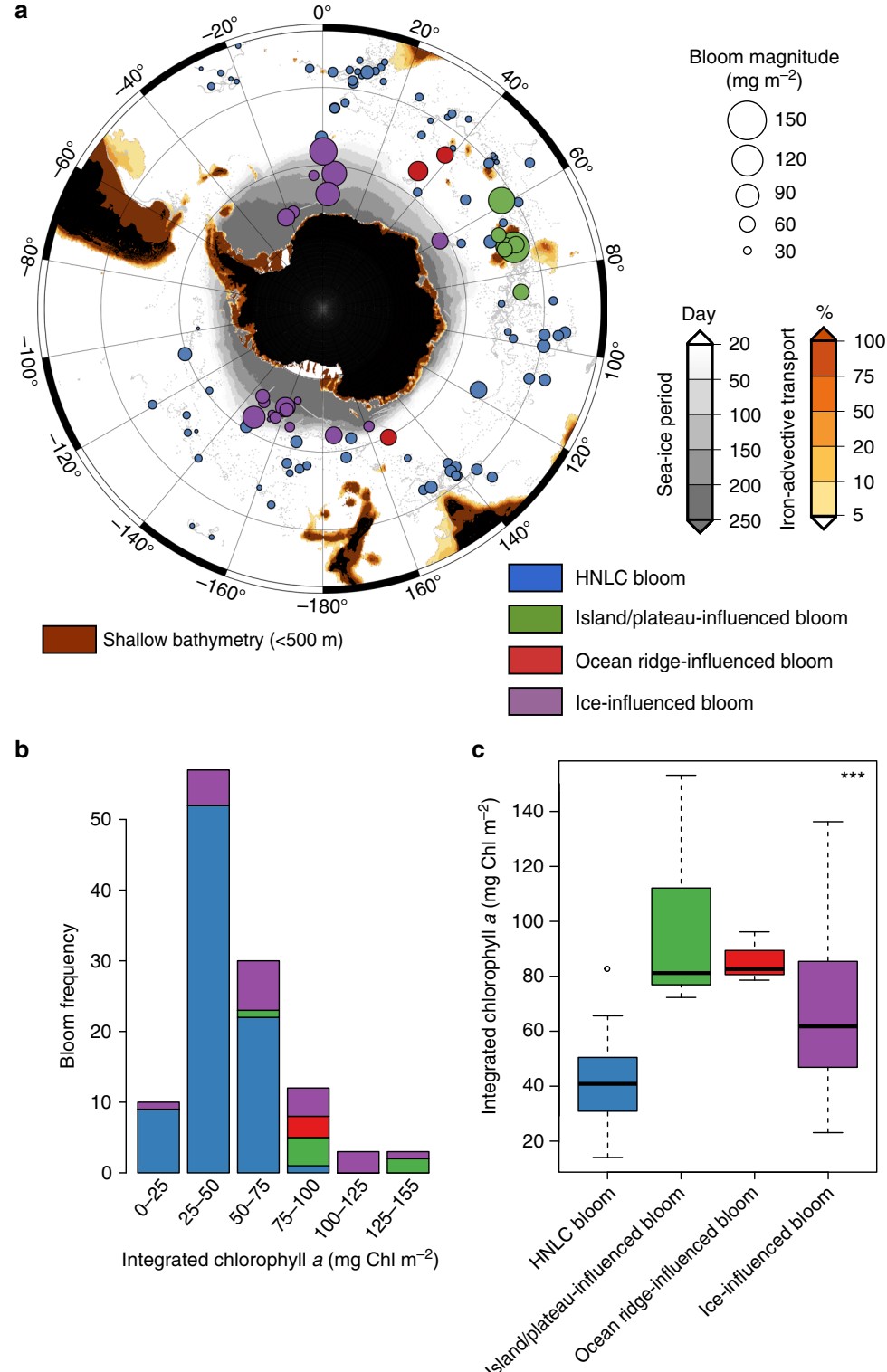

**Fig. 1** Phytoplankton bloom distribution, type and biomass in the Southern Ocean. Map (**a**) of the different bloom types (i.e., blue circles: HNLC; green circle: island/plateau-influenced; red circle: ocean ridge-influenced; purple circle: ice-influenced) sampled. The magnitude of the bloom (i.e., the maximum depth-integrated biomass) is related to the size of the colored circles. The gray dots indicate the individual float profiles. The red, orange, and gray zones are, respectively, shallow areas (>500 m), areas with downstream iron delivery (%; percent of iron remaining in a water parcel after scavenging relative to its initial concentration in shallow areas based on the Lagrangian modeling of horizontal iron delivery), and areas characterized by a seasonal sea ice cover. Histograms (**b**) of the frequency of and boxplot (**c**) according to the bloom type are displayed in relation to the bloom magnitude. In **c**, the top and bottom limits of each box are the 25th and 75th percentiles, respectively. The lines extending above and below each box, i.e., whiskers, represent the full range of non-outlier observations for each variable beyond the quartile range. The results of the Kruskal–Wallis H test are shown in panel **c** and depict regions with statistically significant differences between the magnitudes of the bloom at the 95 % level (p < 0.05). Asterisks (***) denote highly significant results (p < 0.0001)

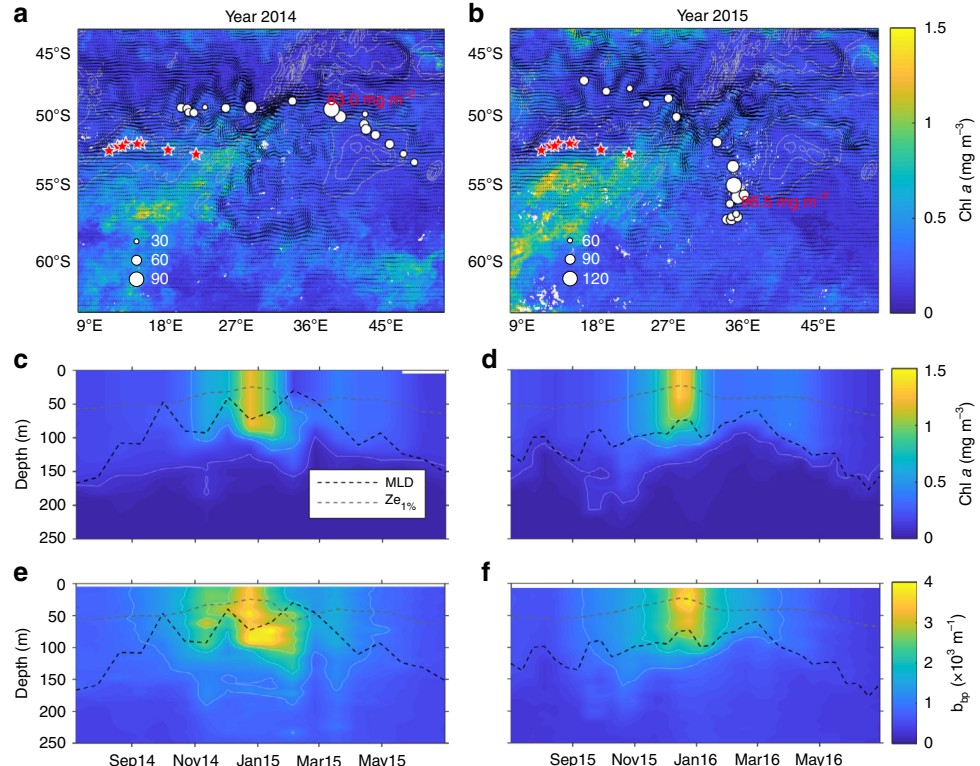

**Fig. 2** Massive phytoplankton blooms stimulated by upwelled hydrothermally influenced deep waters along the Southwest Indian Ridge (SWIR). Maps (**a** and **b**) of the SWIR in the Indian sector of the Southern Ocean and float trajectories. The maximum depth-integrated biomass (mg Chl m$^{-2}$) is depicted according the size of the circles. Satellite-derived surface chlorophyll *a* climatologies (8-days GLOBcolour composite products) were retrieved from November to January **a** 2014–2015 and **b** 2015–2016. Black arrows correspond to altimetry-derived geostrophic velocities (AVISO MADT daily product) averaged over the same period. Gray lines represent the 2000, 3000 and 4000 isobaths. Time series of the 0–250 m vertical distribution of chlorophyll *a* (**c** and **d**) and backscattering (**e** and **f**) for the two BGC-Argo floats (WMO 6901585 and 2902130). The black and gray dashed lines are, respectively, representing the mixed layer depth (determined by a density-derived method with a density threshold of 0.03 kg m$^{-3}$) and the euphotic zone depth (defined as the depth of 1% of surface irradiance according Morel et al.[35]; Eq. 10). The red stars indicate the position of hydrothermal vents from Tao et al.[14]

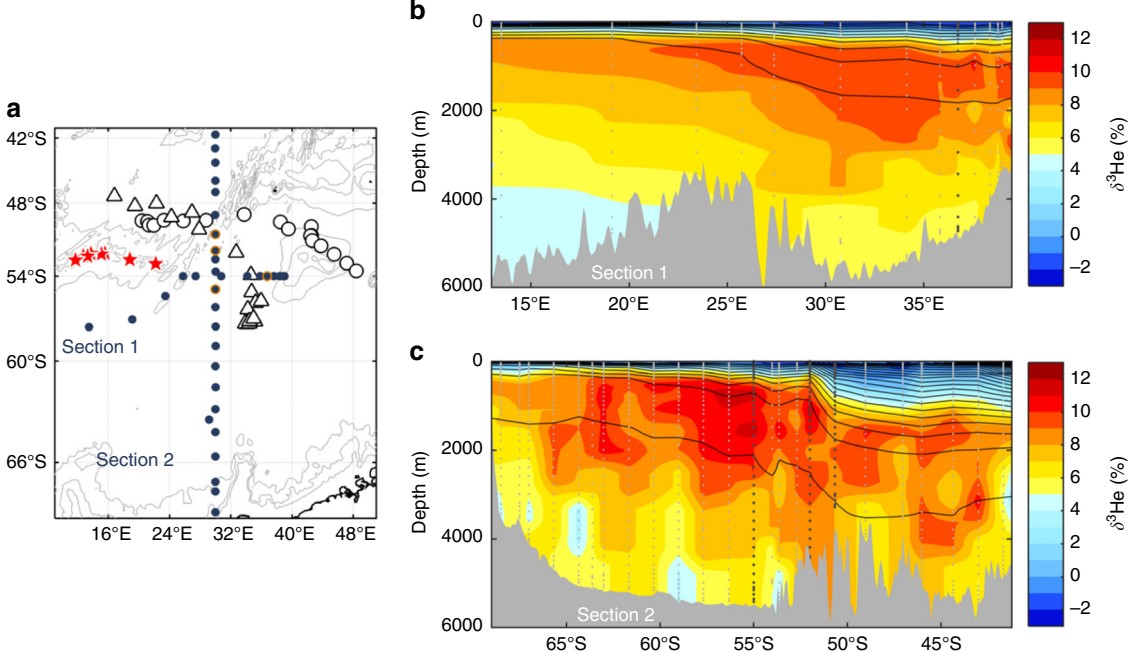

**Fig. 3** Hydrothermally influenced deep waters along the SWIR. Map (**a**) showing the locations of the bathymetry of the SWIR (contour levels: 2000, 3000, and 4000 m), the hydrothermal vents (red stars), the two BGC-Argo floats (black triangle dots: float WMO 6901585 and black circle dots: float WMO 2902130), and of the two sections (filled blue circle dots; **b** and **c**) of interpolated δ$^3$He. Note that all the vertical δ$^3$He profiles, where the surface eddy kinetic energy is high (EKE; >150 cm$^2$ s$^{-2}$), have been highlighted by additional orange circle dots in **a** and by darker gray in **b** and **c**

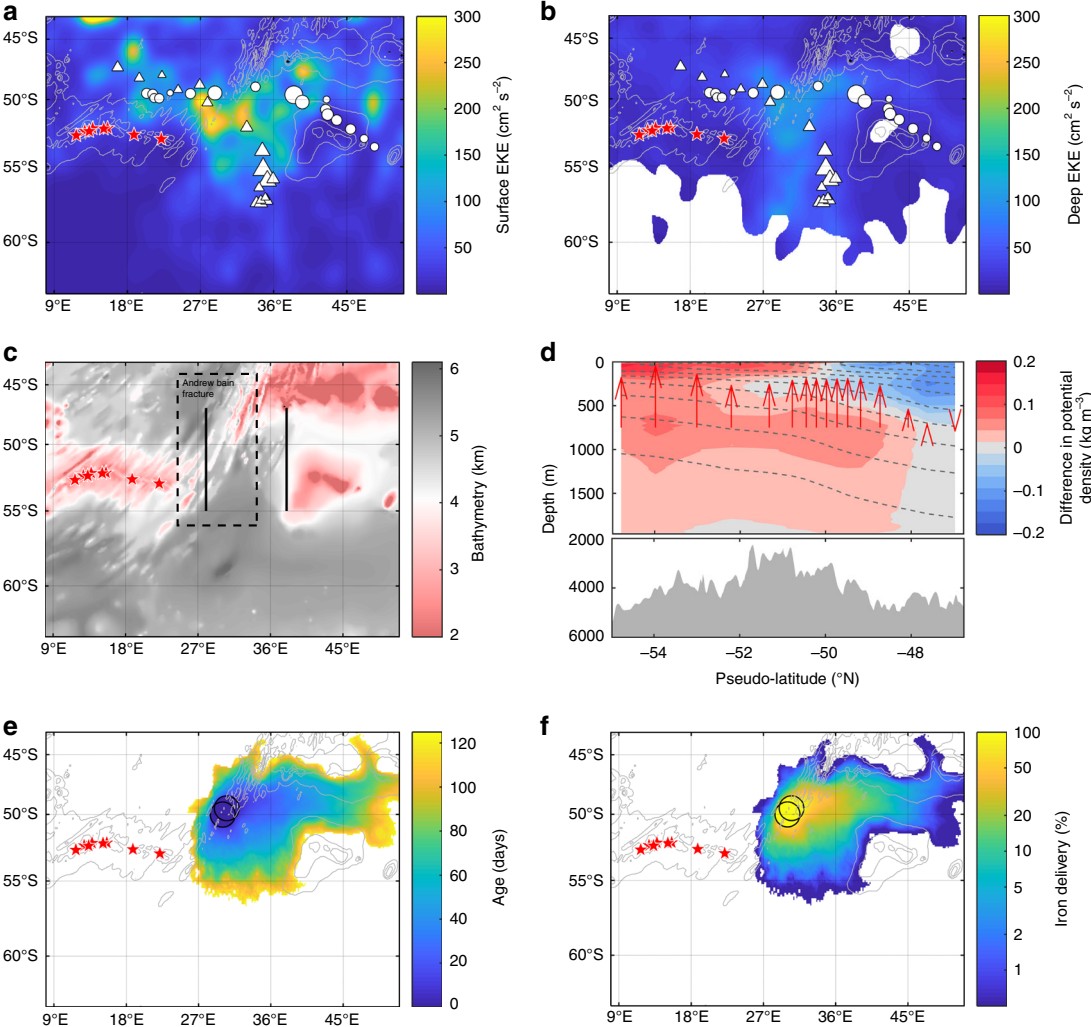

**Fig. 4** Topographically upwelled waters in the vicinity of the Antarctic Circumpolar Current along the SWIR. Maps of the eddy kinetic energy (EKE) at the surface (**a**) and at depth (**b**, approximately 1000 meters). The surface EKE was derived from altimetry-derived velocities (AVISO MADT daily product) over the 2003–2017 period. The deep EKE was calculated from the Argo-derived velocities during their parking depth and available in the ANDRO dataset (2000–2016). The maximum depth-integrated biomass (mg Chl $a$ m$^{-2}$) is also depicted according the size of the dots (triangle: float WMO 6901585 and circle: float WMO 2902130). **c** Maps of bathymetry of the SWIR, the Andrew Bain fracture zone (as indicated by the dashed box; http://www.marineregions.org/gazetteer.php?p=details&id=7253) and the two meridional sections at 28°E and 38°E (between latitude 47–55°S; plain black lines) where the difference in potential density $\Delta\sigma$ shown in panel **d** was determined. **d** Climatological difference of potential density, $\Delta\sigma$, between the two meridional sections at 38°E and 28°E in the upper 2000 m (between latitude 47–55°S). Gray shading represents the mean bottom topography between 28 and 38°E, and the red arrows are provided to show the sense of downstream isopycnal adjustment at 750-m depth. The difference in $\Delta\sigma$ is an alongstream difference across the two sections, which is converted back to latitude for ease of reading (therefore referred to as pseudo-latitude). See the Methods for more details. **e–f** Satellite altimetry-derived Lagrangian modeling of the iron pathways from the departure of the SWIR as shown in **e** age (days since having left the SWIR) and in **f** iron delivery. Black circles in **e–f** indicate the origin of the iron pathways in the surface layer. The red stars in **a–c, e** and **f** are related to the position of hydrothermal vents from Tao et al.[14]. The continuous and dashed gray lines indicate, respectively, the bathymetry (**a**, **b**, **e** and **f**) of the SWIR (contour levels: 2000, 3000, and 4000 m) and the isopycnals (**d**)

suggests that deep, iron rich waters are upwelled directly downstream of the SWIR in the ACC path.

**Horizontal stirring of hydrothermally influenced waters.** To evaluate the advection of upwelled hydrothermally influenced (and likely iron-enriched) waters through horizontal stirring at the surface, we used a Lagrangian satellite altimetry-based method[19]. Briefly, this approach is based on a simple exponential model for iron scavenging over trajectories derived from altimetry and has been extensively validated at similar latitudes in the Kerguelen region[19–21]. As expected, the particles follow the two branches of the ACC and reach the region where we observed

massive bloom 1–2 months later, delivering 10–30% of the surface iron (assuming some particle scavenging[8,19]) contained in waters where the particles originated (Fig. 4e, f). Together, this observationally-based evidence supports a scenario whereby deep-waters are upwelled directly downstream of the SWIR in the ACC path and are then horizontally transported to the area where we observed the large-scale, intense phytoplankton blooms.

**Discussion**
Over the last decade, the conceptual view of the impact of hydrothermal activity on the iron cycle has drastically changed (mostly due to the findings of GEOTRACES expeditions). Model

simulations now suggest that ocean ventilation pathways are potential vectors of spreading hydrothermal trace-elements (including iron) in the SO, hence potentially enhancing the efficiency of the biological pump[10]. Documenting the dynamics of these pathways is particularly challenging because of the remoteness and the extreme conditions of the SO, and the need to be in the right place at the right time. The network of BGC-Argo floats offers the first observational evidence to confirm inferences from models about how hydrothermally stimulated biology is correlated to where [3]He is being upwelled[10], and more importantly, the direct effect of hydrothermal vents on surface biological activity.

Such mechanisms are likely to be more common than we suspected in the SO (as well as in the global ocean), due to the high number of hydrothermal vents (i.e., those identified with many more still undiscovered) and topographically upwelling-favorable features. Here, we captured two massive blooms primarily supported by iron hydrothermal origin. Note that a fraction of hydrothermal iron reported here could also be transported from other basins and remote hydrothermal vents, given evidence suggesting that hydrothermal iron is largely stabilized and so may have a long residence time[4,10]. These results possibly suggest that the other blooms thought to be linked to other iron sources may in fact be due to hydrothermal activity. For example, we suspect that an additional large bloom (maximum depth-integrated biomass of 79.0 mg Chl $a$ m$^{-2}$) in proximity to the sea-ice and an active hydrothermal vent in the northwest Ross Sea (AAR KR2[15]), may be influenced by hydrothermal iron (Fig. 1, red dot).

The implications of such hot spots of biological activity supported by hydrothermal iron are highly significant, by potentially supporting marine ecosystems and sequestering carbon in the SO under the appropriate physical regimes. Traditionally, we assumed that SO phytoplankton blooms were being supported either from continental margin or sea-ice derived iron, but here we demonstrate that hydrothermalism is one additional important forcing for phytoplankton blooms, when associated with the right physics. In summary, a circumpolar analysis is clearly needed to evaluate the overall impact of hydrothermal activity on the carbon cycle in the SO, which appears to trigger local hotspots of enhanced biological pump activity and increase its potential as a sink for atmospheric $CO_2$.

## Methods

**Satellite-derived products**. A satellite-derived Level-3 data set of Chl $a$ concentration (mg m$^{-3}$) was obtained from the European Space Agency's GlobColour project (http://www.globcolour.info). The 8-day composite Chl $a$ concentrations using standard Case 1 water algorithms were used (i.e., OC4Me for Medium-Resolution Imaging Spectrometer, and OC3v5 for Moderate Resolution Imaging Spectroradiometer/Visible Infrared Imaging Radiometer Suite sensors). The altimetry-derived geostrophic velocities (AVISO MADT daily product) were produced by CLS/AVISO (Collecte Localization Satellites/Archiving, Validation, and Interpretation of Satellite Oceanographic data), with the support from the CNES (Centre National d'Etudes Spatiales; http://www.aviso.altimetry.fr/duacs/).

**BGC-Argo network**. The BGC-Argo dataset used in the present study is publicly available at ftp://ftp.ifremer.fr/ifremer/argo/dac/coriolis/, an Argo Global Data Assembly Center. This dataset represents an international initiative by compiling 132 BGC-Argo floats (a total of 14,415 stations) from AOML-NOAA, BODC, CORIOLIS, CSIRO, and INCOIS. It covers the time period from July 2010 to June 2017 and involves a variety of float platforms (NKE's PROVOR, Webb's APEX, and SeaBird's NAVIS) performing different missions (in dive depths, frequency of profiling, and data acquisition) and equipped with different sensors for measuring Chl $a$ and backscattering at 700 nm (Seabird MCOMS, WETLabs Eco-Triplet, and Eco-FLBB).

A quality control procedure was achieved on the CTD[22], Chl $a$[23] and backscattering[24] data. Fluorescence data were corrected for non-photochemical quenching on daytime profiles following the method of Xing et al.[25] as follows: the maximum Chl $a$ value above the mixed layer depth (MLD), defined as a density difference of 0.03 kg m$^{-3}$ from a reference value at 10 m, is extrapolated toward the surface. Fluorescence data are then converted to Chl $a$ concentration (mg m$^{-3}$) by

first applying the calibration (dark value and slope) and then multiplying by a SO-specific correction factor.

To decide which calibration factor to apply, we carried out a robust analysis by applying the radiometric method of Xing et al.[6] to retrieve $F_{490}$ (a refined calibration factor with respect to factory calibration) based on all available BGC-floats equipped with a downward irradiance sensor at 490 nm (OC4 radiometer, Satlantic). We allocated the various BGC-Argo profiles according to SO provinces to detect any potential intra-regional variability in $F_{490}$. The average $F_{490}$ within provinces ranges between 0.26 and 0.33 (Supplementary Fig. 1, analysis of 3321 profiles), which translates into an overestimation factor of 3 to 4 with respect to the factory calibration. This value is actually lower than overestimates derived from HPLC Chl $a$[26,27]. We note here that these HPLC-based estimates (1) are relevant to a spatio-temporal domain restricted to the float deployment (the estimated correction factor might change as environment and community composition change during the float journey) and (2) present a large (yet unexplained) variability. Here, we use a conservative value of 0.3 for $F_{490}$ (corresponding to a factory calibration overestimation of 3.3). This value has the advantage of integrating a broad spatio-temporal domain (e.g., winter conditions) and a large dataset for the estimation of this correction (more than 3300 profiles).

**ANDRO dataset**. The ANDRO atlas ASCII file (available at http://www.coriolis.eu.org/) contains the float parking pressure (actually a representative parking pressure which is generally an average of the measured pressures during float drift at depth) and temperature, deep and surface displacements, and associated times, deep and surface associated velocities with their estimated errors (see Ollitrault and Rannou[28]). ANDRO data originate from AOML, Coriolis, JMA, CSIRO, BODC, MEDS, INCOIS, KORDI, KMA, and CSIO and represent a total of 6271 floats contributing to 612,462 displacements.

Each float cycle (deep displacement between two profiles) provides an estimate of the zonal and meridional current velocities at their drifting depth (mostly around 1000 m). From 2002 to 2016, around 21,300 cycles were available close to the SWIR (Supplementary Fig. 2). These velocities were binned into 1° by 1° boxes and then averaged in space and time. Only the box containing more than 5 data points were kept. Standard deviation of the zonal (u') and meridional (v') velocity components in each box were used to calculate the mean deep eddy kinetic energy (EKE) as follows:

$$\overline{\text{EKE}} = \frac{1}{2}\left(u'^2 + v'^2\right) \tag{1}$$

where the overbar denotes the time average over the whole period (2002–2016). EKE was then interpolated on a finer grid by a Gaussian correlation function, weighted by the local number of data, with a decorrelation radius of 100 km.

**Helium dataset**. The helium data were extracted from the GLobal Ocean Data Analysis Project (GLODAP) Version 2, a cooperative effort to coordinate global synthesis projects funded through NOAA/DOE and NSF as part of the Joint Global Ocean Flux Study–Synthesis and Modeling Project (JGOFS-SMP)[29]. The GLODAP Version 2 data product (available at https://www.glodap.info) is composed of data from 724 scientific cruises covering the global ocean. Here, two different expeditions (06AQ19960317; March 23–31, 1996 on the Polarstern) and (35MF19960220; February 28–March 25, 1996 on the Marion Dufresne) were used to generate the two sections of δ[3]He.

**Isopycnals between meridional section**. A combination of hydrographic profiles from different sources are used to construct a 3-D climatology of potential density in the region 0–55°E and 55–45°S, with a half degree horizontal resolution, and a 25 m vertical resolution.

We use three distinct sources of observations to maximize the number of profiles. The first set of observations is conductivity-temperature-depth (CTD) data from ship-recorded observations during the period 1906–2016 from the NOAA World Ocean Database (https://www.nodc.noaa.gov/OC5/SELECT/dbsearch/dbsearch.html). We only use profiles that have a quality control flag of 1, containing information on their position, date, temperature, and salinity. The second set of observations we use is float observations from the Argo international program. The Argo float profiles of pressure, salinity, and temperature used in this study were gathered in the period 2002–2016. They provide temperature and salinity between 0 and 2000 m. We only use profiles that have a quality control flag of 1, and contain information on their position, date, temperature, and salinity. As a final data set, we use profiles derived from the animal-borne sensor program MEOP (http://www.meop.net/; Treasure et al.[30]). Similar to the other datasets, we only use profiles with control flag of 1, and that contains position, date, temperature, and salinity. Altogether, we gathered 33096 profiles in the region 0–55°E–55–45°S.

From this dataset, we computed potential density for each 25 m vertical interval between the sea surface and 2000 m. Then, for each interval, we produced maps of climatological fields of potential density using an Optimal Interpolation procedure. The Optimal Interpolation and gridding method are described in detail in Schmidtko et al.[31]. As a brief summary, we interpolate onto a 0.5° grid in the region 0–55°E–55–45°S. We used a 550 km isotropic decorrelation scale, incorporating an anisotropic isobath-following component using a "Fast Marching" algorithm, as

well as front-sharpening components. In addition, recent data are emphasized in the mapping, which produces a climatology typical of the years 2000–2010 (see Schmidtko et al.[31] for more details on the mapping).

Two vertical sections of density are extracted at 28°E and 38°E from the produced climatology: $\sigma_{28°}(P,lat)$ and $\sigma_{38°}(P,lat)$. Because the ACC is not entirely zonal, comparing the density structure of these two sections as a zonal difference would compare density structure from south and north of given ACC fronts. Instead, we determined how the density structure differed between 28°E to 38°E but alongstream, i.e., following the ACC structures. For that, we use the dynamical height (dh) provided by AVISO for the period 1993–2012 (http://www.aviso.altimetry.fr/) at these two sections to convert $\sigma_{28°}(P,lat)$ and $\sigma_{38°}(P,lat)$ into $\sigma_{28°}(P,dh)$ and $\sigma_{38°}(P,dh)$. Because fronts and jets tend to follow individual contours of dynamical height (e.g., Sokolov and Rintoul[32]; Sallée et al.[33]), comparing these two sections in dynamic height coordinate ensures an alongstream comparison, i.e., dynamically consistent with regards to the ACC. We therefore produce a difference section: $\Delta\sigma(P,dh)$, and using the mean dynamic height in the sector 28–38°E, we produce a mean relationship between dynamic height and latitude: $\widetilde{lat} = f(dh)$, where $\widetilde{lat}$ is referred to as pseudo-latitude, which we use to produce $\Delta\sigma(P,\widetilde{lat})$.

**Lagrangian modeling of horizontal iron delivery**. An advection scheme based on altimetry was used here to estimate iron delivery due to horizontal stirring from (1) shallow bathymetry (<500 m; as shown in Fig. 1 and in Ardyna et al.[8]) and from (2) the initial location of the iron pathways in the surface layer along the SWIR (Fig. 4e, f). According to the analysis on vertical divergence (Fig. 4d), the origin of the iron pathways in the surface layer is located in the area of vertical displacement of isopycnals, suggesting the upwelling on the SWIR corresponds to the Andrew Bain fracture zone (see Fig. 4c). This feature is located between (24.5°E–56°S) and (34.2°E–44°S) (http://www.ngdc.noaa.gov/gazetteer/). Thus, the enrichment of the two BGC-Argo floats was identified in the region where the floats crossed this geological structure with a high isopycnal adjustment (Fig. 4d). This area has been represented by two overlapping disks centered in (30°E–50°S and 30.5°E–49.5°S), and with a 1° radius, as shown in Fig. 4e, f. We note that another possible upwelling region may occur north or south to this area, according to the analysis of vertical displacement of isopycnals (Fig. 4d).

The advection scheme seeds each location of the study region with a spatial resolution of 1/4° (Fig. 1) and 1/8° (Fig. 4e–f). Trajectories are derived from surface velocities by applying a Runge–Kutta fourth-order scheme with a time step of 6 h, in which velocity fields have been linearly interpolated in both space and time. The advection scheme then finds the particle's most recent contact with an iron source (shallow bathymetry and upwelled input along the SWIR) and provides the time at which the contact took place. The iron content of each particle that was in contact with a potential source of iron was estimated with an exponential scavenging relation. This relation reproduces the decreasing concentration of bioavailable iron along the trajectory after the contact with the iron source. This approach was initially developed for predicting the development of the Kerguelen phytoplanktonic plume[19] and thereafter extended to the entire Southern Ocean[8]. The model has been calibrated and validated in the Crozet and Kerguelen regions by combining satellite data (altimetry and ocean color), lithogenic isotopes, iron measurements, and drifters[19–21] (see d'Ovidio et al.[19] for further details). Here the advection scheme is applied to the period 2010–2015 and for the planktonic bloom season, November to March, in order to obtain a mean climatological signal.

**Bloom characterization**. To determine the bloom magnitude, each float time series was divided into individual annual cycles, starting on 1 July. Cycles that do not cover the theoretical bloom period (from early November to late February the next year) with at least eight float profiles were discarded. For each remaining cycle, float profiles were binned into a 20-day period corresponding to the decorrelation scale of float Chl a records[34]. The magnitude of the bloom was then computed as the maximum in integrated Chl a biomass from the surface down to either the MLD or the euphotic depth, whichever was deeper. The euphotic depth is the depth at which light is 1% of its surface value, based on the surface Chl a according to Morel et al.[35].

## Data availability
All the data used in this research are freely available and may be downloaded through the links detailed in the Methods section.

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

## Acknowledgements

M.A. was supported by a CNES (Centre National d'Etudes Spatiales) Postdoctoral Fellowship and by a European Union's Horizon 2020 Marie Sklodowska-Curie grant (no. 746748). This work represents a contribution to the remOcean project (REMotely sensed biogeochemical cycles in the OCEAN, GA 246777) funded by the European Research Council and to the project SOCLIM (Southern Ocean and climate) supported by the French research program LEFE-CYBER of INSU-CNRS, the Climate Initiative of the foundation BNP Paribas, the French polar institute (IPEV), the CNES Tosca/OSTST (project LAECOS) and Sorbonne Université. J.-B.S. was funded by the European Research Council (ERC) under the European Union's Horizon 2020 research and innovation program (grant agreement 637770). A.T. was funded by the European Research Council (ERC) under the European Union's Horizon 2020 research and innovation program (grant agreement 724289). We would like to thank the international agencies and programs, including the U.S. National Science Foundation's Southern Ocean Carbon and Climate Observations and Modeling (SOCCOM) for freely providing access to their float data.

## Author contributions

M.A., L.L., S.S., F.d'O., J.-B.S., M.R., A.T. conducted the data analysis. M.A., J.-B.S., A.T., F.d'O., K.R.A. wrote the manuscript. All authors contributed to the ideas and commented on the manuscript.

## Additional information

**Competing interests:** The authors declare no competing interests.

