## [Peer Review File · Nature Communications]

Reviewer #1 (Remarks to the Author):

One of the most influential findings in ocean science during the last century is that biological production in extensive areas of the global ocean (~30%) is limited by the nutrient iron, which is present only in trace amounts. The Southern Ocean is one such region and has little to no terrestrial inputs (riverine or Aeolian dust) of iron. Instead, seasonal convective mixing, resuspension from shallow bathymetric features, and melting of sea ice provide some relief to chronic iron limitation, but a vast reservoir of unused macro-nutrients (e.g., nitrate and phosphate) go unused. As a result, primary production and the potential of the biological pump is limited in its efficiency to sequester CO₂ from the surface ocean. In this work, the authors provide a previously undocumented and surprising source for iron to the surface ocean – the seafloor. The authors put forward evidence for hydrothermally sourced iron from the deep ocean (>3000m) fueling biological production in upper ~100 m of the surface ocean.

These conclusions are extremely original and of broad interest. The story presented is very complete from end-to-end from and documents hydrothermal influence on overlying waters (via He tracers), vertical transport pathways (via Argo and altimetry), and bloom characterization (via BGC-Argo floats). The authors have done a very thorough job in this sense and I find no fault with the logic. However, I do point out one seemingly trivial, yet possibly critical problem. The authors evaluate the significance of the two blooms using water-column integrated chlorophyll inventories estimated from fluorescence sensors. They apply well-documented corrections to these data to recover Chl-a concentration. One of these corrections is for a recently characterized bias in the fluorescence sensors that causes a ~2x overestimation in the estimated Chl-a concentration globally (Roeseler et al., 2017). However, in that same paper Roeseler et al. show that in the Southern Ocean this bias is more like 6.4 (see their Figure 2 and Table 1). These results have been further reported and elaborated upon in the works of Johnson et al. (2017) and Haentjens et al. (2017). The important point here is that the Chl-a concentrations that are reported in this manuscript are likely a factor of 3x too high! Reducing the hydrothermally-induced bloom Chl-a inventories by 3x results in integrated Chl-a of ~43-50 mg Chl m⁻². These values are comparable to winter/spring values in the Sargasso Sea at the BATS site. I think the authors would agree these would certainly not qualify as “massive” blooms, as stated in the title. If true, this does not negate the mechanism, nor potential for the pathway from seafloor iron to surface bloom. It does however, significantly lessen the impact of the results and calls into question suitability of publication in Nature, as opposed to a disciplinary journal. I welcome a rebuttal, as I this is one of the more interesting reports I have seen in the literature lately and recalls a more old-fashioned discovery-based science reporting which is often lost in today’s world.

On a related note, why not try and convert the corresponding backscattering measurements bbp into POC and estimate the relative change before and at the peak of the Chl signal. Is it smaller than seen in the Chl record? Going further, backscattering could also be converted into phytoplankton biomass (following Behrenfeld et al., 2005 and related papers) in order to determine how much of the bloom in Chl is attributable to an upregulation in physiology versus simply biomass accumulation? Just a thought to help address my first comment above.

Haentjens, N., Boss, E. and Talley, L. D. (2017), Revisiting Ocean Color algorithms for chlorophyll a and particulate organic carbon in the Southern Ocean using biogeochemical floats, *J. Geophys. Res. Oceans*, doi:10.1002/2017JC012844.

Johnson, K. S., et al. (2017), Biogeochemical sensor performance in the SOCCOM profiling float array, *J. Geophys. Res. Oceans*, 122, doi:10.1002/2017JC012838.

Roesler, et al. (2017), Recommendations for obtaining unbiased chlorophyll estimates from in situ chlorophyll fluorometers: A global analysis of WET Labs ECO sensors, *Limnol. Oceanogr. Methods*, 15, 572–585, doi:10.1002/lom3.10185.

Reviewer #2 (Remarks to the Author):

Review of the manuscript by Dr Ardyna and co-workers entitled “Hydrothermal vents trigger massive phytoplankton blooms in the Southern Ocean”, submitted to *Nature Communication*

I have organized my review along the referee guidelines. My summarized recommendation is “accept after minor revision.”

- What are the major claims of the paper?

This paper supports the hypothesis of ocean modeling work and empirical studies on hydrothermal contribution to the ocean biogeochemical cycle by using satellite data and profiling floats, showing that hydrothermally influenced water from the South-Western Indian Ridge SWIR reaches the photic zone by upwelling and causes phytoplankton blooms. This is especially relevant in the nutrient-starved Southern Ocean, where this process may act as a local hotspot for surface bioproductivity and as a sink for atmospheric carbon dioxide.

- Are the claims novel? If not, please identify the major papers that compromise novelty

The claims are not really new, and there are many citations in this manuscript that refer to the former hypothesis. However, so far, not much evidence has been given for the role of hydrothermal nutrient input into the oceans, and hence, this work provides new important evidence.

- Will the paper be of interest to others in the field?

This paper will generally be interesting for people working on ocean biogeochemistry, the role of iron and other hydrothermally derived elements to the ocean, scientists focusing on the Southern Ocean, and physical oceanographers.

- Will the paper influence thinking in the field?

This paper gives further support for the hypothesis that hydrothermalism plays a more important role in ocean biogeochemistry and surface bioproductivity than previously thought. It is one further step in moving forward in this direction when considering the elemental budgets in the ocean and their role for climate processes.

- Are the claims convincing? If not, what further evidence is needed?

I am not an expert in the field of working with satellite data and data from Argo floats, however, the reasoning occurs convincing to me.

- Are there other experiments that would strengthen the paper further? How much would they improve it, and how difficult are they likely to be?

It would have been nice if the theory of upwelling of hydrothermally derived iron supporting the phytoplankton blooms had been supported by field work tracing the iron flux from the SWIR to the areas where the blooms were observed. However, this was clearly beyond the possibilities of this study but could be a way to go in future studies.

- Are the claims appropriately discussed in the context of previous literature?

I am more an expert on field work and analytical investigations of ocean biogeochemistry and not exactly on the type of work carried out in this project, hence, it is a bit difficult for me to answer this question. Previous literature is discussed and the present work connected to the previous findings and theories.

- Is the manuscript clearly written? If not, how could it be made more accessible?

Yes, the manuscript is concise and clearly written.

- Could the manuscript be shortened to aid communication of the most important findings?

No, the manuscript is already quite short and all information is necessary.

- Have the authors done themselves justice without overselling their claims?

In my opinion, yes.

- Have they been fair in their treatment of previous literature?

Yes, they have given credit to those studies that have brought up the theory of significant hydrothermal contributions to surface bioproductivity in the Southern Ocean.

- Have they provided sufficient methodological detail that the experiments could be reproduced?

Since I am not an expert in this type of work, I cannot judge this question.

- Is the statistical analysis of the data sound?

Not applicable

- Should the authors be asked to provide further data or methodological information to help others replicate their work? (Such data might include source code for modelling studies, detailed protocols or mathematical derivations).

Methods information appears detailed and adequate. However, as said above, I am not an expert in this kind of work.

I have some additional minor comments.

I think the term eddy kinetic energy (EKE) has not been explained in the main text but only the abbreviation is used.

Figure 1, Caption: % percent of iron remaining in a water parcel after scavenging ...; I do not understand how these values were determined or where they were taken from, respectively.

Figure 2, c to f: it seems to me like the labels on the right side on the y axis are not correct or at least they do not fit the assignment to c, d, e and f. If c and e is the chlorophyll distribution, they both must have the unit "Chla" and not "b" (for e). The red stars are a bit difficult to identify in the dark blue background.

Revised paper of Ardyna et al. “Hydrothermal vents trigger massive phytoplankton blooms in the Southern Ocean”

We thank the reviewers for their kind comments and useful suggestions. The main concern from Reviewer #1 has been fully addressed in the manuscript and detailed below, as well as the minor comments of Reviewer #2. Our responses to their comments are below in bold, non-italicized text. The reviewers’ comments are italicized.

Reviewer 1:

One of the most influential findings in ocean science during the last century is that biological production in extensive areas of the global ocean (~30%) is limited by the nutrient iron, which is present only in trace amounts. The Southern Ocean is one such region and has little to no terrestrial inputs (riverine or Aeolian dust) of iron. Instead, seasonal convective mixing, resuspension from shallow bathymetric features, and melting of sea ice provide some relief to chronic iron limitation, but a vast reservoir of unused macro-nutrients (e.g., nitrate and phosphate) go unused. As a result, primary production and the potential of the biological pump is limited in its efficiency to sequester CO₂ from the surface ocean. In this work, the authors provide a previously undocumented and surprising source for iron to the surface ocean – the seafloor. The authors put forward evidence for hydrothermally sourced iron from the deep ocean (>3000m) fueling biological production in upper ~100 m of the surface ocean. These conclusions are extremely original and of broad interest. The story presented is very complete from end-to-end from and documents hydrothermal influence on overlying waters (via He tracers), vertical transport pathways (via Argo and altimetry), and bloom characterization (via BGC-Argo floats). The authors have done a very thorough job in this sense and I find no fault with the logic.

We completely agree with the reviewer.

However, I do point out one seemingly trivial, yet possibly critical problem. The authors evaluate the significance of the two blooms using water-column integrated chlorophyll inventories estimated from fluorescence sensors. They apply well-documented corrections to these data to recover Chl-a concentration. One of these corrections is for a recently characterized bias in the fluorescence sensors that causes a ~2x overestimation in the estimated Chl-a concentration globally (Roesler et al., 2017). However, in that same paper Roesler et al. show that in the Southern Ocean this bias is more like 6.4 (see their Figure 2 and Table 1). These results have been further reported and elaborated upon in the works of Johnson et al. (2017) and Haentjens et al. (2017). The important point here is that the Chl-a concentrations that are reported in this manuscript are likely a factor of 3x too high! Reducing the hydrothermally-induced bloom Chl-a inventories by 3x results in integrated Chl-a of ~43-50 mg Chl m⁻². These values are comparable to winter/spring values in the Sargasso Sea at the BATS site. I think the authors would agree these would certainly not qualify as “massive” blooms, as stated in the title. If true, this does not negate the mechanism, nor potential for the pathway from seafloor iron to surface bloom. It does however, significantly lessen the impact of the results and calls into question suitability of publication in Nature, as opposed to a disciplinary journal. I welcome a rebuttal, as I think this is one of the more interesting reports I have seen in the literature lately and recalls a more old-fashioned discovery-based science reporting which is often lost in today’s world.

As noted by the reviewer #1, the fluorescence calibration may seem trivial, but this is actually not the case. It still remains an important issue, which has only recently been

emphasized due to the development of the BGC-Argo fleet (e.g. Haëntjens et al. (2017); Roesler et al. (2017)). This issue is especially important for the Southern Ocean where the Wetlabs fluorometer is likely to overestimate Chl *a* concentration. It is indeed true that there is no clear consensus on the calibration factor to be used in the SO. To decide which calibration factor to apply, we carried out a robust analysis by applying the radiometric method of Xing et al. (2011) to retrieve F_{490} (a refined calibration factor with respect to factory calibration) based on all available BGC-floats equipped with a downward irradiance sensor at 490 nm (OC4 radiometer, Satlantic). We allocated the various BGC-Argo profiles according to SO provinces to detect any potential intra-regional variability in F_{490} . The average F_{490} within provinces ranges between 0.26 to 0.33 (Figure 1, analysis of 3321 profiles), which translates into an overestimation factor of 3 to 4 with respect to the factory calibration. This value is actually lower than overestimates derived from HPLC Chl *a* (Haëntjens et al. 2017; Roesler et al. 2017). We note here that these HPLC-based estimations (1) are relevant to a spatio-temporal domain restricted to the float deployment (the estimated correction factor might change as environment and community composition change during the float journey) and (2) present a large (yet unexplained) variability. Here we use a conservative value of 0.3 for F_{490} (corresponding to a factory calibration overestimation of 3.3). This value has the advantage of integrating a broad spatio-temporal domain (e.g. winter conditions) and a large dataset for the estimation of this correction (more than 3300 profiles). This SO-specific fluorescence correction is now described in the SI (see lines 468-487) and mentioned in the main text (see lines 201-202).

Fig. 1: Box and whisker plots of the F_{490} factor calculated for each SO province (based on the temperature profiles following Pollard et al. (2002) and Swart et al. (2010); STZ: Subtropical zone, SAZ: Subantarctic zone, PFZ: Polar Frontal Zone, AAZ: Antarctic Zone, SACCZ: the zone south of the Antarctic Circumpolar Current). The median value is specified.

Fortunately, applying this new calibration correction does not change the main conclusion of our manuscript about the importance of these hydrothermally-derived blooms (even if they are less massive as we previously calculated, as well as the entire SO). By compiling the largest *in situ* circumpolar dataset of phytoplankton blooms, we still demonstrate that these hydrothermally-influenced blooms (maximum depth-integrated biomass of 96.5 and 83.0 mg Chl *a* m⁻²) are still “massive” for the SO, similar to the most highly productive

areas in the SO, e.g., the iron-enriched waters downstream of the Crozet and Kerguelen Plateaus (mean 98.1 mg Chl *a* m⁻²) and in proximity to the sea-ice edge (mean 70.0 mg Chl *a* m⁻²), and more than twice that of HNLC waters of the SO (mean 42.0 mg Chl *a* m⁻²). Based on this large-scale analysis, we are still convinced that these hydrothermally-derived blooms are particularly crucial as hot-spots in the biogeochemistry of the SO surrounded by highly oligotrophic HNLC waters (comparable to Sargasso Sea waters).

On a related note, why not try and convert the corresponding backscattering measurements (b_{bp}) into POC and estimate the relative change before and at the peak of the Chl signal. Is it smaller than seen in the Chl record? Going further, backscattering could also be converted into phytoplankton biomass (following Behrenfeld et al., 2005 and related papers) in order to determine how much of the bloom in Chl is attributable to an upregulation in physiology versus simply biomass accumulation? Just a thought to help address my first comment above.

Analyzing the net biomass growth rates (based on both chl *a* and b_{bp}) is particularly interesting to understand the mechanisms behind bloom initiation (in fact, we are currently working on a manuscript on this specific question). Here, it is beyond the scope of our letter, focused (as mentioned by Reviewer #1) on describing a complete story documenting hydrothermal influence on overlying waters (via He tracers), vertical transport pathways (via Argo and altimetry), and bloom characterization (via BGC-Argo floats). And as a note, calculating phytoplankton carbon biomass will not provide additional insights on phytoplankton dynamics, due to its linear relationship to b_{bp} (from Boss and Behrenfeld (2010)).

Reviewer 2:

What are the major claims of the paper?

This paper supports the hypothesis of ocean modeling work and empirical studies on hydrothermal contribution to the ocean biogeochemical cycle by using satellite data and profiling floats, showing that hydrothermally influenced water from the South-Western Indian Ridge SWIR reaches the photic zone by upwelling and causes phytoplankton blooms. This is especially relevant in the nutrient-starved Southern Ocean, where this process may act as a local hotspot for surface bioproductivity and as a sink for atmospheric carbon dioxide.

We thank the reviewer for this insight. No response from us is required.

- *Are the claims novel? If not, please identify the major papers that compromise novelty. The claims are not really new, and there are many citations in this manuscript that refer to the former hypothesis. However, so far, not much evidence has been given for the role of hydrothermal nutrient input into the oceans, and hence, this work provides new important evidence.*

We thank the reviewer for this insight. No response from us is required.

- *Will the paper be of interest to others in the field?*
This paper will generally be interesting for people working on ocean biogeochemistry, the role of iron and other hydrothermally derived elements to the ocean, scientists focusing on the Southern Ocean, and physical oceanographers.
- *Will the paper influence thinking in the field?*
This paper gives further support for the hypothesis that hydrothermalism plays a more

important role in ocean biogeochemistry and surface bioproductivity than previously thought. It is one further step in moving forward in this direction when considering the elemental budgets in the ocean and their role for climate processes.

We thank the reviewer for this insight. No response from us is required.

- *Are the claims convincing? If not, what further evidence is needed?*
I am not an expert in the field of working with satellite data and data from Argo floats, however, the reasoning occurs convincing to me.

We thank the reviewer for this insight. No response from us is required.

- *Are there other experiments that would strengthen the paper further? How much would they improve it, and how difficult are they likely to be?*
It would have been nice if the theory of upwelling of hydrothermally derived iron supporting the phytoplankton blooms had been supported by field work tracing the iron flux from the SWIR to the areas where the blooms were observed. However, this was clearly beyond the possibilities of this study but could be a way to go in future studies.

We thank the reviewer for this insight. No response from us is required.

- *Are the claims appropriately discussed in the context of previous literature?*
I am more an expert on field work and analytical investigations of ocean biogeochemistry and not exactly on the type of work carried out in this project, hence, it is a bit difficult for me to answer this question. Previous literature is discussed and the present work connected to the previous findings and theories.

We thank the reviewer for this insight. No response from us is required.

- *Is the manuscript clearly written? If not, how could it be made more accessible?*
Yes, the manuscript is concise and clearly written.
- *Could the manuscript be shortened to aid communication of the most important findings?*

We thank the reviewer for this insight. No response from us is required.

No, the manuscript is already quite short and all information is necessary.

We thank the reviewer for this insight. No response from us is required.

- *Have the authors done themselves justice without overselling their claims?*
In my opinion, yes.

We thank the reviewer for this insight. No response from us is required.

- *Have they been fair in their treatment of previous literature?*

We thank the reviewer for this insight. No response from us is required.

Yes, they have given credit to those studies that have brought up the theory of significant

hydrothermal contributions to surface bioproductivity in the Southern Ocean.

We thank the reviewer for this insight. No response from us is required.

- *Have they provided sufficient methodological detail that the experiments could be reproduced?*

Since I am not an expert in this type of work, I cannot judge this question.

We thank the reviewer for this insight. No response from us is required.

- *Is the statistical analysis of the data sound?*

Not applicable

We thank the reviewer for this insight. No response from us is required.

- *Should the authors be asked to provide further data or methodological information to help others replicate their work? (Such data might include source code for modelling studies, detailed protocols or mathematical derivations).*

Methods information appears detailed and adequate. However, as said above, I am not an expert in this kind of work.

We thank the reviewer for this insight. No response from us is required.

I think the term eddy kinetic energy (EKE) has not been explained in the main text but only the abbreviation is used.

Agreed. We now define eddy kinetic energy in the manuscript (see line 130)

Figure 1, Caption: % percent of iron remaining in a water parcel after scavenging ...; I do not understand how these values were determined or where they were taken from, respectively.

Agreed. We improved the legend and the methods (see lines 233-245 and 351-352). The “% percent of iron remaining in a water parcel after scavenging” is based on the Lagrangian modeling of iron delivery.

Figure 2, c to f: it seems to me like the labels on the right side on the y axis are not correct or at least they do not fit the assignment to c, d, e and f. If c and e is the chlorophyll distribution, they both must have the unit “Chla” and not “b” (for e). The red stars are a bit difficult to identify in the dark blue background.

The figure 2 has been modified (as well as Figures 3 and 4) by improving the visualization of the hydrothermal vents (by changing the contour of the stars) and by correcting the labelling of the figures (see lines 368).

References

- Boss, E., and M. Behrenfeld. 2010. In situ evaluation of the initiation of the North Atlantic phytoplankton bloom. *Geophys. Res. Lett.* **37**: L18603.
- Haëntjens, N., E. Boss, and L. D. Talley. 2017. Revisiting Ocean Color algorithms for chlorophyll a and particulate organic carbon in the Southern Ocean using biogeochemical floats. *Journal of Geophysical Research: Oceans*.
- Pollard, R. T., M. I. Lucas, and J. F. Read. 2002. Physical controls on biogeochemical zonation in the Southern Ocean. *Deep Sea Res. Pt. 2* **49**: 3289-3305.
- Roesler, C., J. Uitz, H. Claustre and others 2017. Recommendations for obtaining unbiased chlorophyll estimates from in situ chlorophyll fluorometers: A global analysis of WET Labs ECO sensors. *Limnol. Oceanogr. Methods*.
- Swart, S., S. Speich, I. J. Ansorge, and J. R. E. Lutjeharms. 2010. An altimetry-based gravest empirical mode south of Africa: 1. Development and validation. *J. Geophys. Res.* **115**: 2156-2202.
- Xing, X., A. Morel, H. Claustre, D. Antoine, F. D'ortenzio, A. Poteau, and A. Mignot. 2011. Combined processing and mutual interpretation of radiometry and fluorimetry from autonomous profiling Bio-Argo floats: Chlorophyll a retrieval. *J. Geophys. Res.* **116**: C06020.

Reviewer #1 (Remarks to the Author):

This is a re-review, so the attached comments are fairly short and only focus on previous concerns that were raised.

Reviewer #3 (Remarks to the Author):

This paper presents new observations from BGC-Argo floats of large phytoplankton blooms downstream of the Southwest Indian Ridge, in a region of the Southern Ocean that is typically thought of as a high nutrient low chlorophyll region. The authors argue that that the bloom are driven by upwelling of hydrothermally-sourced iron from vents located nearby. The paper is well written and figures are clear, and the topic is of great interest to others in the community and in the wider field.

However, I have two major comments on the manuscript. First, some of the methods are unclear as it stands or are not described in enough detail, making it difficult to evaluate the robustness of the results. Second, I don't think the conclusion that the blooms are sources solely by nearby hydrothermal vents is clearly supported by the results presented here. In spite of this, I think this paper is of significant importance, but requires some modification of the conclusions.

It may be my own misinterpretation, but the method and data used to determine the density difference between two meridional sections is unclear as it stands. Additionally, there are very few details included on the Lagrangian analysis, and it would be really beneficial to include more details on this, rather than relying entirely on referencing earlier papers. See more detailed comments on this below.

2. I think the authors have shown convincingly that the source of the iron must be upwelling from the subsurface ocean, but to me I am not totally convinced from the current evidence whether much is directly sourced from the nearby hydrothermal vents, or how much could be simply from enhanced upwelling/vertical mixing of higher background subsurface iron as a result of flow/topography interactions (e.g. Tagliabue et al. 2014). Clearly hydrothermal iron sources contribute a large amount to the enhanced subsurface iron in the Southern Ocean, but the direct link to the nearby hydrothermal vents is tenuous. The authors use Lagrangian particle tracking and altimetry-derived velocities to show iron advection can reach the float locations from a given location where iron is assumed to have upwelled to the surface.

However, this method using altimetry is able to show surface horizontal advection of iron, from a location where the two float tracks intersect. There is no way currently how and where the deep iron is upwelled from the vents to the surface. The choice of release location for the Lagrangian tracking seems arbitrarily related to the crossing of the float trajectories, and it is unclear to me that

the area of vertical isopycnal displacements (which is broad) necessarily means that iron-rich deep water first enters the surface layer in this specific location. This location is also substantially north of the hydrothermal events, and it is not clear how the hydrothermally sourced iron is advected and upwelled to the surface at this northerly location.

Given the limited subsurface data available, answering this question fully with currently available observational data would be extremely challenging. Instead it would likely require conducting Lagrangian experiments with a 3D velocity field from a high-resolution ocean model, which is clearly too large an undertaking for this current work. I think modification of the conclusions to say that hydrothermal iron is likely a large contribution of iron to the blooms observed, rather than the sole contributor, this will be more in line with the results.

Minor comments

Line 69: seems weird that it would extend west... unless source is upstream? I think perhaps you mean eastward here (see comments further below as well).

Line 73: need to define SO acronym

Line 97-99: Any references for this sentence? You could refer to iron advection from shallow bathymetry shown in colour on Fig 1a here, as evidence for these blooms being away from iron sources from shallow bathymetry.

Line 101-104: what depth are these hydrothermal vents at? This isn't stated anywhere (although it can sort of be deduced from the topography contours in Figure 2a and 3a) and would be very helpful for knowing where to expect injection of high $\delta^{13}\text{C}$ waters in the observed sections, and to understand the vertical extent of upwelling required to bring this source iron to the surface ocean.

Line 106-107: all three of these things need to be shown clearly for the conclusions to be robust

Line 114-116: A constant relationship between iron and $\delta^{13}\text{C}$ is an important assumption, how important is this to the interpretation of your results?

Line 117-119: Are there no direct observations of iron that can confirm iron-rich waters in this region?

Line 119-121: It would be nice to relate the maxima in $\delta^{13}\text{C}$ section 2 (Fig. 3c) at 51-52S and 55-56S to ACC jets/branches of high EKE that are at similar latitudes at 30E (Fig. 4b).

Line 124: I think you mean east for downstream, not west!

Line 135: There are more references you could include here that relate upwelling to high EKE in this region, including Tamsitt et al. 2017, which is referenced earlier in the paper.

Line 141- 142: May be worth mentioning here that the deep EKE is derived from Argo trajectory-based velocities so the reader doesn't have to refer to the Figure caption.

Line 142: How are the isopycnal displacements calculated? It also doesn't say anywhere what data is used to calculate the isopycnal displacement (see further comments on Figure 4 below).

Line 150: Is this a tracer release? Or particles? I think it is particles, used to represent the tracer but it is important that the language is clear here.

Line 152: How is this percentage of total iron supply determined? Do you mean that 10-30% of the iron initially injected to the surface at the particle release location reached the location of the blooms? Please clarify.

Line 153: I wouldn't call the Lagrangian particle results observational evidence, but rather observationally-based.

Line 169: As discussed above, based on my current understanding of the results presented (although I may be misunderstanding some of the results as the methods are not detailed), I don't think the conclusion that the blooms are 'supported solely by iron of thermal origin' is convincingly presented here.

Line 171-174: It would be useful to mention this other large bloom in the Ross Sea that is potentially associated with hydrothermal iron sources earlier in the manuscript, when Figure 1 is first described. Otherwise it may leave the reader wondering why this similar bloom is not discussed.

Line 219-224: What is the cruise information (section name, dates etc) of the two sections shown of helium data presented here?

Line 240: I don't think the Andrew Bain fracture zone is a well known feature, it would be useful to include the lat/lon of this feature or label it on one of the maps.

Line 241-242: Latitude should be 50S and 49.5S, not -50N etc. It would be very useful to indicate these overlapping disks on the figure, perhaps on Figure 4e.

Line 282: Beaulieu et al. reference appears to be missing information?

Line 234-237: It is necessary to include at least some details of the Lagrangian analysis used. Is this using the daily altimetry-based velocity product, and what is the timestep for the Lagrangian integration (and are the velocities interpolated in time/space)? I know the methods are described in more detail in D'Ovidio et al. 2015, which is referenced in this section but it would be very helpful to include some basic details here.

Line 349-351: Are the grey dots representing individual float profiles? It would be good to add what these are.

Line 361: 'figure' should be 'panel'

Line 372: 'BCG-Argo' should be 'BGC-Argo'

Line 379: What are the contour levels for the bathymetry?

Line 380: 'et' should be 'and'. What are the black lines on b and c? Are they isopycnals?

Line 388-390: From the explanation given here for the difference in potential density, it is not clear to me how this was calculated, and using which data.

Line 390: -47- -55N should be 47-55S.

Figures

Figure 3a: It would be nice to add the two float tracks to this map.

Figure 4:

d) It is difficult to determine how the density difference as a function of depth and pseudo latitude was calculated for Figure 4d from the information provided in the text and figure caption. Two meridional sections are defined to show where the density difference was calculated, but it unclear which density data is used for this (is it from the floats? Or from the hydrographic sections shown in Figure 3?), nor is it explained how the 'pseudo-latitude' is determined.

e) As mentioned earlier, it would be great to show the locations of particle release here.

Supplementary Information

You need to include specific references to each of the supplementary Figure and text items in the main body of the article.

Line 462-463: In the EKE equation the squares need to be superscript, and there is currently no overbar, which needs to be added.

Revised paper of Ardyna et al. "Hydrothermal vents trigger massive phytoplankton blooms in the Southern Ocean"

We thank all the reviewers, in particular Toby Westberry for his re-review and Reviewer #3 for the new comments and useful suggestions for improving our manuscript. The comments from the reviewers have been addressed in the manuscript and detailed below. Our responses to their comments are below in bold, non-italicized text. The reviewers' comments are italicized.

Reviewer #1:

As documented in my previous review, this was already a nice manuscript. I only raised one major concern regarding the calibration of the BGC-Argo fluorescence sensors and their well documented bias (Haentjens et al. 2017; Johnson et al., 2017). The concern was that if this bias was taken into account, it would significantly reduce the recovered chlorophyll concentration (by ~3x). In other words, it would make the "massive" blooms seem not so massive. The authors responded by acknowledging the uncertainty in the calibration factors and carried out their own analysis using the BGC-Argo floats to characterize this calibration problem using the method of Xing et al. (2011). While I personally maintain that the results from Haentjens et al. (2017) are preferred and are based on direct measurements of the property being estimated (chlorophyll concentration), I am OK with their approach. It has been published, accepted by the community (for the most part), and widely applied in the literature. However, it might be noted that their method provides a potential upper bound for the recovered chlorophyll and thus "bloom size" when compared with results using the calibration bias of Haentjens et al. (2017).

We appreciate that the reviewer agrees with the appropriateness of our novel SO-specific calibration using the method of Xing et al. (2011). We should also mention that independent HPLC measurements (as in Haentjens et al., 2017) at time of float deployment in the Indian sector of the Southern Ocean have revealed (data not shown) overestimation similar to that derived from Xing et al. (2011) so that the 3X can be definitively considered as an upper limit.

Also, my previous suggestion to use the concurrent backscattering retrievals to estimate "bloom size" in terms of POC was a bit misunderstood. My point was simply that if there is significant variability in the conversion of fluorescence to chlorophyll, perhaps looking at the relative change in another geophysical property (POC) would be instructive. That is, if the presumed bloom is an accumulation of phytoplankton biomass, then POC should scale with chlorophyll concentration and the relative changes in both properties should be comparable. I realize there are caveats with other components of POC and of course, the conversion of backscattering to POC itself. Nevertheless, it seemed like an easy thing to do, particularly considering that the basic BGC-Argo floats give us two pieces of "bio"-information, chlorophyll fluorescence (which was used), and particulate backscattering (which was ignored).

As suggested by the reviewer, we added a new section in the Supplemental Information focused on the use of the particulate backscattering. We performed the same analysis as

shown in Figure 1 to confirm the quantitative importance of the blooms in the vicinity of the SWIR (please see the new analysis in the Supplementary Result 2.1).

Reviewer #3:

This paper presents new observations from BGC-Argo floats of large phytoplankton blooms downstream of the Southwest Indian Ridge, in a region of the Southern Ocean that is typically thought of as a high nutrient low chlorophyll region. The authors argue that that the bloom is driven by upwelling of hydrothermally-sourced iron from vents located nearby. The paper is well written and figures are clear, and the topic is of great interest to others in the community and in the wider field.

We thank the reviewer for this insight. No response from us is required.

However, I have two major comments on the manuscript. First, some of the methods are unclear as it stands or are not described in enough detail, making it difficult to evaluate the robustness of the results. Second, I don't think the conclusion that the blooms are sources solely by nearby hydrothermal vents is clearly supported by the results presented here. In spite of this, I think this paper is of significant importance, but requires some modification of the conclusions.

- 1. It may be my own misinterpretation, but the method and data used to determine the density difference between two meridional sections is unclear as it stands. Additionally, there are very few details included on the Lagrangian analysis, and it would be really beneficial to include more details on this, rather than relying entirely on referencing earlier papers. See more detailed comments on this below.*

Agreed. The methods have been improved by adding a new section describing the calculation of values shown in Figure 4d (see the Supplementary Methods 1.2) and the legend of Figure 4. Additionally, more information is given in the Lagrangian modeling section (see lines 240-268).

- 2. I think the authors have shown convincingly that the source of the iron must be upwelling from the subsurface ocean, but to me I am not totally convinced from the current evidence whether much is directly sourced from the nearby hydrothermal vents, or how much could be simply from enhanced upwelling/vertical mixing of higher background subsurface iron as a result of flow/topography interactions (e.g. Tagliabue et al. 2014). Clearly hydrothermal iron sources contribute a large amount to the enhanced subsurface iron in the Southern Ocean, but the direct link to the nearby hydrothermal vents is tenuous. The authors use Lagrangian particle tracking and altimetry-derived velocities to show iron advection can reach the float locations from a given location where iron is assumed to have upwelled to the surface.*

One of the highlights of this study is the large coverage of BGC-Argo floats and the ability to capture a significant number of blooms in unique and diverse environmental conditions. Given the magnitude of the two specific blooms we describe, comparable to those near the sea ice or with shallow bathymetry, and twice as high as those observed in the HNLC waters,

we were convinced that an additional iron source was playing an important role in driving these blooms. Far from any potential and known iron source, we found that hydrothermal vents were the only plausible iron contributor.

Interestingly, of the numerous blooms observed in HNLC waters, some of them are found in regions with significant flow/topography interactions with no significant impacts on bloom magnitude. In fact, the BGC-Argo network provides a wealth of information, on which we are still working on complementary aspects (including flow/topography interactions) that drive the productivity, phenology and structure of phytoplankton in the SO.

However, this method using altimetry is able to show surface horizontal advection of iron, from a location where the two float tracks intersect. There is no way currently how and where the deep iron is upwelled from the vents to the surface. The choice of release location for the Lagrangian tracking seems arbitrarily related to the crossing of the float trajectories, and it is unclear to me that the area of vertical isopycnal displacements (which is broad) necessarily means that iron-rich deep water first enters the surface layer in this specific location. This location is also substantially north of the hydrothermal events, and it is not clear how the hydrothermally sourced iron is advected and upwelled to the surface at this northerly location. Given the limited subsurface data available, answering this question fully with currently available observational data would be extremely challenging. Instead it would likely require conducting Lagrangian experiments with a 3D velocity field from a high-resolution ocean model, which is clearly too large an undertaking for this current work. I think modification of the conclusions to say that hydrothermal iron is likely a large contribution of iron to the blooms observed, rather than the sole contributor, this will be more in line with the results.

We agree with the reviewer that it would be more appropriate to modify the conclusions by saying that hydrothermal vents likely contributed a large fraction of iron utilized by the blooms observed, rather than the sole contributor (see line 174). For the rest of the questions linked to the methods and the Lagrangian modeling, we hope that they are now fully addressed in the improved sections describing the methods.

Minor comments:

Line 69: seems weird that it would extend west... unless source is upstream? I think perhaps you mean eastward here (see comments further below as well).

In Resing et al. (2015), the hydrothermal plume from the southern east Pacific rise vent field indeed extends westward across the Pacific basin.

Line 73: need to define SO acronym

Agreed. We now define Southern Ocean in the manuscript (see line 73).

Line 97-99: Any references for this sentence? You could refer to iron advection from shallow bathymetry shown in colour on Fig 1a here, as evidence for these blooms being away from iron sources from shallow bathymetry.

Agreed. We referred to the Fig 1a for blooms away from both shallow bathymetry and sea-ice, and to Jickells and Moore (2015) for atmospheric dust deposition (see line 99).

Line 101-104: what depth are these hydrothermal vents at? This isn't stated anywhere (although it can sort of be deduced from the topography contours in Figure 2a and 3a) and would be very helpful for knowing where to expect injection of high $\delta^3\text{He}$ waters in the observed sections, and to understand the vertical extent of upwelling required to bring this source iron to the surface ocean.

Agreed. This new information has been included in the main text (see lines 103). For information, these hydrothermal vents are located from 3517 to 4170 meters deep.

Line 106-107: all three of these things need to be shown clearly for the conclusions to be robust.

Agreed. As mentioned in the first general comment, the description of the methods has been improved to support the robustness of our results.

Line 114-116: A constant relationship between iron and $\delta^3\text{He}$ is an important assumption, how important is this to the interpretation of your results?

While it has been long noted that there is likely a variable Fe/He flux away from hydrothermal systems (Tagliabue et al., 2010), this aspect is not crucial for this study. Instead, here we use the helium signal as a fingerprint for the influence of hydrothermally sourced water. In all mid ocean ridge GEOTRACES sections to date, elevated helium signals have been associated with positive dFe anomalies, even distally from source.

Line 117-119: Are there no direct observations of iron that can confirm iron-rich waters in this region?

Unfortunately, no direct observations of iron are available in this region, this is the reason why we used a proxy of hydrothermal vents (i.e., the $\delta^3\text{He}$).

Line 119-121: It would be nice to relate the maxima in $\delta^3\text{He}$ section 2 (Fig. 3c) at 51-52S and 55-56S to ACC jets/branches of high EKE that are at similar latitudes at 30E (Fig. 4b).

Agreed. As proposed, we modified Figure 3 to include the location where we found high EKE. Even by combining $\delta^3\text{He}$ and EKE (which do not cover the same period), it appeared to be informative and shows a good correspondence between high $\delta^3\text{He}$ and high EKE.

Line 124: I think you mean east for downstream, not west!

Agreed. We changed “west” to “east” (see line 126).

Line 135: There are more references you could include here that relate upwelling to high EKE in this region, including Tamsitt et al. 2017, which is referenced earlier in the paper.

Agreed. We included here the reference Tamsitt et al. 2017.

Line 141- 142: May be worth mentioning here that the deep EKE is derived from Argo trajectory-based velocities so the reader doesn't have to refer to the Figure caption.

Agreed. The sentence has been modified as the reviewer suggested (see lines 144-145).

Line 142: How are the isopycnal displacements calculated? It also doesn't say anywhere what data is used to calculate the isopycnal displacement (see further comments on Figure 4 below).

The methods have been improved by adding a new section that describes how we calculate the Figure 4d (see the Supplementary Methods 1.2) and the legend of Figure 4 has been improved.

Line 150: Is this a tracer release? Or particles? I think it is particles, used to represent the tracer but it is important that the language is clear here.

Agreed. We changed the word “tracer” to “particles” (see line 154).

Line 152: How is this percentage of total iron supply determined? Do you mean that 10-30% of the iron initially injected to the surface at the particle release location reached the location of the blooms? Please clarify.

Agreed. We clarified this sentence in the methods (see lines 154-157).

Line 153: I wouldn't call the Lagrangian particle results observational evidence, but rather observationally-based.

Agreed. We modified the sentence as the reviewer suggested (see line 157).

Line 169: As discussed above, based on my current understanding of the results presented (although I may be misunderstanding some of the results as the methods are not detailed), I don't think the conclusion that the blooms are 'supported solely by iron of thermal origin' is convincingly presented here.

As mentioned in the second general comment, we agree with the reviewer that it would be more appropriate to slightly modify the conclusions by saying that hydrothermal iron is likely a large contributor of iron to the blooms observed, rather than the sole contributor (see line 174).

Line 171-174: It would be useful to mention this other large bloom in the Ross Sea that is potentially associated with hydrothermal iron sources earlier in the manuscript, when Figure 1 is first described. Otherwise it may leave the reader wondering why this similar bloom is not discussed.

In the present study, we intentionally did not focus on the potential hydrothermally-influenced bloom in the Ross Sea, but just mention it as an avenue of future investigations. The reason is that contrary to the two blooms in the vicinity of the SWIR, in the Ross Sea

there might be a strong influence of iron-enrichment from melting sea-ice and upwelling on the shelf. It would be impossible with BGC-Argo floats to disentangle the respective influences of all these potential iron contributors. We rather here prefer to focus on the SWIR for clarity, and to hopefully convincingly demonstrate that hydrothermal iron is likely a large iron contributor to the blooms observed without ambiguity.

Line 219-224: What is the cruise information (section name, dates etc) of the two sections shown of helium data presented here?

We added the cruise information used here (see lines 228-229)

Line 240: I don't think the Andrew Bain fracture zone is a well known feature, it would be useful to include the lat/lon of this feature or label it on one of the maps.

Agreed. We added the Andrew Bain fracture zone in the Figure 4c and we define it in the methods (see lines 245-247).

Line 241-242: Latitude should be 50S and 49.5S, not -50N etc. It would be very useful to indicate these overlapping disks on the figure, perhaps on Figure 4e.

Agreed. We added these overlapping disks on the Figure 4e and 4f.

Line 282: Beaulieu et al. reference appears to be missing information?

We corrected the Beaulieu et al. reference (see lines 307-309).

Line 234-237: It is necessary to include at least some details of the Lagrangian analysis used. Is this using the daily altimetry-based velocity product, and what is the timestep for the Lagrangian integration (and are the velocities interpolated in time/space)? I know the methods are described in more detail in D'Ovidio et al. 2015, which is referenced in this section but it would be very helpful to include some basic details here.

Agreed. The method section on the Lagrangian modeling has been improved (see lines 240-268).

Line 349-351: Are the grey dots representing individual float profiles? It would be good to add what these are.

Agreed. This information has been added to the legend (see lines 378-379).

Line 361: 'figure' should be 'panel'

Agreed. We changed 'figure' to 'panel' (see line 387)

Line 372: 'BCG-Argo' should be 'BGC-Argo'

Agreed. We changed 'BCG-Argo' to 'BGC-Argo' (see line 398)

Line 379: What are the contour levels for the bathymetry?

The contour levels for the bathymetry are 2000, 3000 and 4000 meters. This information has been added in the legend (see lines 405, 431-433).

Line 380: 'et' should be 'and'. What are the black lines on b and c? Are they isopycnals?

Agreed. We changed 'et' to 'and' and the legend of Figure 4 has been improved indicating the meaning of the black lines on b and c (please refer to the new legend of Figure 4).

Line 388-390: From the explanation given here for the difference in potential density, it is not clear to me how this was calculated, and using which data.

Agreed. The methods have been improved by adding a new section that describes how we calculated the information shown in Figure 4d (see the Supplementary Methods 1.2)

Line 390: -47- -55N should be 47-55S.

Agreed. We changed '47- -55N' to '47-55S' (see line 420).

Figures

Figure 3a: It would be nice to add the two float tracks to this map.

Agreed. The float tracks have been added to the figure.

Figure 4: d) It is difficult to determine how the density difference as a function of depth and pseudo latitude was calculated for Figure 4d from the information provided in the text and figure caption. Two meridional sections are defined to show where the density difference was calculated, but it unclear which density data is used for this (is it from the floats? Or from the hydrographic sections shown in Figure 3?), nor is it explained how the 'pseudo-latitude' is determined.

Agreed. The methods have been improved by adding a new section that describes how we calculated the values shown in Figure 4d (see the Supplementary Methods 1.2)

e) As mentioned earlier, it would be great to show the locations of particle release here.

Agreed. The figure has been modified as the reviewer suggested.

Supplementary Information

You need to include specific references to each of the supplementary Figure and text items in the main body of the article.

Agreed. We specifically related all the supplementary sections in the main text of the article.

Line 462-463: In the EKE equation the squares need to be superscript, and there is currently no overbar, which needs to be added.

Agreed. The equation has been corrected (see line 503).

References

Jickells, T. D. & Moore, C. M. The importance of atmospheric deposition for ocean productivity. *Annu. Rev. Ecol. Evol. Syst.* **46**, 481-501, doi:10.1146/annurev-ecolsys-112414-054118 (2015).

Resing, J. A. et al. Basin-scale transport of hydrothermal dissolved metals across the South Pacific Ocean. *Nature* **523**, 200-203, doi:10.1038/nature14577 (2015).

Tagliabue, A. et al. Hydrothermal contribution to the oceanic dissolved iron inventory. *Nature Geosci* **3**, 252-256, doi:10.1038/ngeo818 (2010).

Reviewer #3 (Remarks to the Author):

I thank the authors for their careful consideration of the review comments, and I think they the addition of more detailed methods has greatly improved the clarity of the paper and ability to interpret the results and figures. I am satisfied that they have responded to all of my minor comments and I think the paper is now close to being ready to publish in Nature Communications. However, I still think that there is some subtleties in the conclusions that are not fully supported by the observations and analysis that the authors have not sufficiently addressed in the first review. I therefore recommend minor revisions.

I have attempted to outline this more specifically in detail below and have made a suggestion for how to address these points.

The authors have shown from convincingly from the observations that the observed blooms are predominantly fuelled by iron of hydrothermal origin (shown in the $\delta^{13}\text{C}$ sections)

But, to me it still not shown convincingly to me from the results whether

a) the hydrothermal iron upwells directly from the specific hydrothermal vents on the SWIR (stars in Fig. 2a) to the surface to fuel the blooms via enhanced upwelling in high EKE regions downstream of topography.

or

b) there is elevated iron in the deep ocean in the region, that originates from hydrothermal inputs both locally in the Southern Ocean and hydrothermal iron transported into the Southern Ocean from other basins (e.g. Tagliabue et al. 2010, 2016), that reaches the surface by enhanced upwelling in high EKE regions downstream of topography. In this case, the physical mechanism upwelling the iron is the same as a), but the iron is not tied to a local hydrothermal source.

For a) in particular, there is no mention in the paper for how the iron injected at 4500-5000m depth where the vents are located on the SWIR ends up with a maximum iron concentration at 1000-1500 m (based on the $\delta^{13}\text{C}$ section 2 in Fig. 3) 5 degrees longitude downstream. Based on what is presented, it seems feasible to me that a significant fraction of this iron could instead have been transported to the region from more remote hydrothermal sources (given evidence suggesting that hydrothermal iron is largely stabilised and so may have a long residence time).

I understand that with existing observations determining how much of the hydrothermal iron upwelled to the surface ocean downstream of the SWIR is contributed locally vs remotely would be very difficult to answer and so may require analysis in a model but is clearly beyond the scope of this current work. As such, I suggest the authors include some discussion of the potential role of local vs remote hydrothermal sources.

Revised paper of Ardyna et al. "Hydrothermal vents trigger massive phytoplankton blooms in the Southern Ocean"

We thank the Reviewer #3 for his last comment for improving our manuscript. This comment has been addressed in the manuscript and detailed below.

Reviewer #3 (Remarks to the Author):

I thank the authors for their careful consideration of the review comments, and I think they the addition of more detailed methods has greatly improved the clarity of the paper and ability to interpret the results and figures. I am satisfied that they have responded to all of my minor comments and I think the paper is now close to being ready to publish in Nature Communications. However, I still think that there is some subtleties in the conclusions that are not fully supported by the observations and analysis that the authors have not sufficiently addressed in the first review. I therefore recommend minor revisions.

I have attempted to outline this more specifically in detail below and have made a suggestion for how to address these points.

The authors have shown from convincingly from the observations that the observed blooms are predominantly fuelled by iron of hydrothermal origin (shown in the $\delta^{13}\text{C}$ sections)

But, to me it still not shown convincingly to me from the results whether

a) the hydrothermal iron upwells directly from the specific hydrothermal vents on the SWIR (stars in Fig. 2a) to the surface to fuel the blooms via enhanced upwelling in high EKE regions downstream of topography.

or

b) there is elevated iron in the deep ocean in the region, that originates from hydrothermal inputs both locally in the Southern Ocean and hydrothermal iron transported into the Southern Ocean from other basins (e.g. Tagliabue et al. 2010, 2016), that reaches the surface by enhanced upwelling in high EKE regions downstream of topography. In this case, the physical mechanism upwelling the iron is the same as a), but the iron is not tied to a local hydrothermal source.

For a) in particular, there is no mention in the paper for how the iron injected at 4500-5000m depth where the vents are are located on the SWIR ends up with a maximum iron concentration at 1000-1500 m (based on the $\delta^{13}\text{C}$ section 2 in Fig. 3) 5 degrees longitude downstream. Based on what is presented, it seems feasible to me that a significant fraction of this iron could instead have been transported to the region from more remote hydrothermal sources (given evidence suggesting that hydrothermal iron is largely stabilised and so may have a long residence time).

I understand that with existing observations determining how much of the hydrothermal iron upwelled to the surface ocean downstream of the SWIR is contributed locally vs remotely would be very difficult to answer and so may require analysis in a model but is clearly beyond the scope of this current work. As such, I suggest the authors include some discussion of the potential role of local vs remote hydrothermal sources.

As suggested by the reviewer, a new discussion on local *versus* remote hydrothermal sources has been added (see lines 181-184).